# Evaluation of Yellow Fever Virus Infection in *Aedes aegypti* Mosquitoes from Pakistan with Distinct Knockdown Resistance Genotypes

**DOI:** 10.3390/insects16010033

**Published:** 2024-12-31

**Authors:** Carlucio Rocha dos Santos, Caleb Guedes Miranda dos Santos, Dinair Couto-Lima, Bárbara Silva Souza, Rafi Ur Rahman, Marcos Dornelas Ribeiro, José Bento Pereira Lima, Ademir Jesus Martins

**Affiliations:** 1Laboratório de Biologia, Controlee Vigilância de InsetosVetores, Instituto Oswaldo Cruz, FIOCRUZ, Rio de Janeiro 21040-360, RJ, Brazilbarbara.s.souza@hotmail.com (B.S.S.); rafi.rahman@ioc.fiocruz.br (R.U.R.);; 2Instituto de Biologia do Exército, Divisão de Ensino e Pesquisa, Rio de Janeiro 20911-270, RJ, Brazildornelas-ribeiro@hotmail.com (M.D.R.); 3Laboratório de Mosquitos Transmissores de Hematozoários, Instituto Oswaldo Cruz, FIOCRUZ, Rio de Janeiro 21040-360, RJ, Brazil; 4Instituto Nacional de Ciência e Tecnologia (INCT), Universidade Federal do Rio de Janeiro, Rio de Janeiro 21941-902, RJ, Brazil

**Keywords:** mosquito, *Aedes aegypti*, yellow fever, insecticide resistance, *kdr*, Asia, arboviruses, infection

## Abstract

Background: Yellow fever (YF) is a serious disease found in Africa and Latin America that is potentially transmitted by *Aedes aegypti* in urban regions. Surprisingly, YF did not occur in Asia, despite the presence of these mosquitoes. This may be related to environmental factors, genetic aspects of the virus strains and mosquito populations. Moreover, as insecticide resistance mutations, such as *kdr*, influence mosquito physiology, such mutations might affect its infectivity by the yellow fever virus (YFV). Therefore, we examined whether mosquitoes from Pakistan with distinct *kdr* genotypes could be infected and replicate the YF virus. Methods: Female *Aedes aegypti* from Pakistan were exposed to two YFV strains in the laboratory. A laboratory mosquito strain was used for comparison. After two weeks, the mosquitoes were tested to determine if the virus had spread through their bodies. Genetic analysis was performed to identify *kdr* mutations related to insecticide resistance. Results: Pakistani mosquitoes were susceptible to YFV, with infection rates similar to those of laboratory strains. Two *kdr* mutations were detected; however, these mutations did not affect the ability of mosquitoes to become infected with YFV. Conclusions: *Ae. aegypti* mosquitoes from Pakistan can become infected with YFV even if they possess mutations associated with insecticide resistance.

## 1. Introduction

Yellow fever (YF) is an arboviral infection with a wide range of clinical manifestations ranging from asymptomatic cases to severe YF, which may include hemorrhagic conditions and hepatic impairment [1]. It is estimated that there are over 200,000 annual cases, of which 5–10% are severe, with death rates ranging from 20 to 80% [2,3]. Vaccination programs are often hindered by low immunization coverage and failures in social communication, undermining the effectiveness of disease control and highlighting YF as a significant public health concern [4,5]. The yellow fever virus (YFV) has three main transmission cycles involving vectors from *Aedes*, *Haemagogus*, and *Sabethes* genera and vertebrate hosts. These cycles are divided into a sylvatic cycle, in which the circulation between sylvatic vectors, such as the *Haemagogus* and *Sabethes* genera, and non-human primates (NHPs) can accidentally infect humans; the intermediate or savannah cycle between semi-domestic mosquitoes from the *Aedes* genus and humans or NHPs; and the urban cycle, in which humans are the main hosts of urban vectors, such as the *Ae. aegypti* species [6,7]. These cycles are predominantly found in the Americas, with two YFV genotypes circulating in the sylvatic and urban cycles (due to a spillover from the sylvatic cycle), and in Africa, with a more diverse landscape accounting for 90% of the annual global burden of YFV, presenting five YFV genotypes and all three transmission cycles, with the intermediate one usually serving as a bridge for the urban cycle [6,8]. The maintenance of these cycles depends on several factors that ensure the conditions required for transmission between different vectors and hosts [6,9]. These factors are divided into extrinsic factors, related to susceptible host/vector presence and environmental parameters such as temperature, humidity, and food availability, and intrinsic factors, including immune responses, microbiota, and genetic factors, encompassing vector and viral genotypes, which are important determinants of overall vector competence [10,11]. Indeed, an extrinsic factor with a key role in human YFV infection is the presence and distribution of susceptible mosquito vectors, such as the mosquito *Aedes* (Stegomyia) *aegypti* (Linnaeus, 1762), which is considered one of the main species responsible for the urban YF cycle [11]. This mosquito is highly anthropophilic and adapted to anthropogenic habitats, living near human settlements, and is found in greater frequency and abundance in urban environments, predominantly in tropical and subtropical zones [12,13,14].

Despite the high density of *Ae. aegypti* in urban environments inAsia, there have never been records of YF transmission in Asian countries, whereas other *Ae. aegypti* arboviruses, such as dengue and chikungunya, are responsible for epidemics in the Asia-Pacific region as well as in the American and African continents [15,16,17,18]. Several hypotheses have been proposed for the absence of YF in Asia, including neglected surveillance and incorrect disease diagnostics, cross-protection against YFV provided by dengue antibodies, and low or absent vector competence of *Ae. aegypti* populations to YFV in that region [19,20,21,22]. However, natural populations of *Ae*. *aegypti* may present different levels of susceptibility to arbovirus infection as a consequence of the genetic structuring of mechanisms necessary for mosquito infection, replication, and transmission of the virus. Laboratory infection assays demonstrated that populations of *Ae. aegypti* from Asian countries, such as Thailand, Vietnam, and Singapore, were competent to be infected and transmit YFV [16,19,23,24,25,26,27,28].

*Aedes aegypti* is a widespread mosquito species in Pakistan, a densely populated subtropical country in Asia bordering populous nations such as China and India, with a growing urban population and frequent dengue and chikungunya outbreaks, creating all the conditions necessary for potential yellow fever transmission cycles [29,30]. Additionally, the selection of insecticide resistance threatens the control of these mosquito species [31,32]. Pyrethroids are the main class of insecticides employed, and the principal physiological mechanisms selected for resistance to its *knockdown* effect are mutations in the target-site molecule, the voltage-gated sodium channel (*Na_V_*), known as *knockdown* resistance mutations (*kdr*) [33,34]. The emergence and spread of *kdr* in *Ae. aegypti* have converged for mutations in several *Na_V_* sites, both worldwide, such as the F1534C substitution, and regional-specific, such as the V410L and V1016I mutations in the Americas and Africa and the S989P, V1016G, and T1520I mutations in Asia [35,36,37]. In *Ae. aegypti* from Pakistan, at least two *kdr* mutations have been described (T1520I and F1534C), forming two *kdr* haplotypes: T1520+1534C (*kdr* at the 1534 site) and 1520I+1534C (*kdr* at both sites) [32]. Moreover, *kdr* mutations have been associated with negative pleiotropic effects in mosquitoes, resulting in fitness costs, such as prolonged larval development, higher pupal mortality, reduced body weight, and oviposition rates [38,39]. As the presence of *kdr* mutations can alter the normal fitness of the mosquito, we hypothesized that mosquitoes with distinct *kdr* genotypes could present differences in their capacity to be infected and replicate YFV [38,39,40]. In *Anopheles* species, for example, the presence of *kdr* mutation has been correlated with a higher *Plasmodium falciparum* [38,40]. In the present study, we orally administered *Ae. aegypti* from Pakistan, carrying two different *kdr* genotypes with two strains of YFV under laboratory conditions. We then compared the infection and dissemination rates, as well as the viral load.

## 2. Material and Methods

### 2.1. Ethics Statements

The Oswaldo Cruz Institute has received accreditation for carrying out experiments with wild and urban mosquitoes kept as colonies in insectaries. This study was approved by the Institutional Ethics Committee on Animal Use (CEUA IOC-License LO28/18) of the Oswaldo Cruz Institute, FIOCRUZ.

### 2.2. Mosquito Rearing

We used *Ae*. *aegypti* originally from a collection in the Lahore district, Pakistan, and from our constantly reared Rockefeller strain as an internal reference for laboratory assays [32,41]. These colonies were maintained by hatching the eggs in 2 L of dechlorinated water containing 1 g of fish food (Tetramin^®^ granules, Tetra GmbH, Melle, Germany). After 24 h, the larvae were divided into trays (500 larvae per tray) containing 2 L of dechlorinated water and 1 g of fish food, which was added every 48 h until pupation. The pupae were then separated into 50mL plastic cups and transferred to cardboard entomological cages for adult emergence. Adults were maintained under *ad libitum* feeding with 10% sucrose and allowed to copulate. For oviposition, females were starved overnight and fed defibrinated rabbit blood the following day using the artificial feeder Hemotek^®^ (PS-6 System, Discovery Workshops, Accrington, UK). Eggs from the Pakistani F6 and Rockefeller (undetermined generation) generations were used in the assays. The colonies were maintained at LBCVIV, IOC, and FIOCRUZ.

### 2.3. YFV and Infection Procedures

For the laboratory infection assays, we used two distinct South American YFV strains, maintained at LATHEMA, IOC, and FIOCRUZ: PR4408 (PR4408/2008), isolated from an *Alouatta* sp. (howler monkey) in 2008, from lineage 1E (GenBank KY861728), and ES504/2017, isolated from the serum of a non-human primate, *Alouatta guariba clamitans* (GenBank KY88500), modern lineage (sublineage 1E) of South American genotype I [42]. One day before infection, we transferred 50 *Ae*. *aegypti* females, 7-to-9-day-old, to six 300 mL small cages for each viral strain tested. Only water was offered to the mosquitoes for 24 h, followed by a 24 h complete fasting period before infective blood feeding. The mosquitoes were artificially blood-fed with a 1:1 (*v*/*v*) solution of rabbit erythrocytes and L-15 culture medium containing one of the YFV strains (PR4408/2008 or ES504) [43]. The viral titers in the blood meal solution consisted of PR4408/2008–3.3 × 10^5^ and ES504–2.7 × 10^5^, as described elsewhere [43]. Feeding was performed with artificial Hemotek^®^ feeders kept at 37 °C for approximately 2 h in a Biosafety Level 3 insectary (BSL-3) at the Army Biology Institute (IBEx). After blood feeding, the mosquitoes were anesthetized on ice and screened to transfer only the fully engorged females to new cages, kept in Biochemical Oxygen Demand (B.O.D.) incubators at 28 °C and a 12h photoperiod in a BSL-3 insectary at IBEx, with a 10% sucrose solution. After 14 days, mosquitoes were collected and frozen at −70 °C until dissection and nucleotide extraction.

### 2.4. Mosquito Dissection

Mosquitoes were dissected into the head, thorax, and abdomen according to the LBCVIV/LATHEMA protocol using a sterilized scalpel between each dissected mosquito to avoid cross-contamination among samples. Heads and abdomens were used for RNA extraction and infection evaluation, whereas thoraxes were used for DNA extraction and *kdr* genotyping. After dissection, the samples were transferred to microtubes and stored at −70 °C.

### 2.5. RNA Extraction and YFV Detection and Quantification

For RNA extraction, heads and abdomens were thawed and homogenized using a TissueLyser II (Qiagen^®^, Venlo, Limburg, The Netherlands) with the TRIzol (ThermoFisher^®^, Waltham, MA, USA) extraction protocol, as described by the manufacturer. After extraction, RNA concentration was determined using a Qubit 2.0 Fluorometer with the Qubit RNA High Sensitivity Kit (Invitrogen^®^, Waltham, MA, USA), following the manufacturer’s instructions, and the nucleic acid was stored in microtubes at −70 °C until assay. The reactions for YFV detection were performed using the RT-qPCR TaqMan Fast Virus 1-Step Master Mix kit (ThermoFisher^®^, Waltham, MA, USA). Reactions consisted of 1X TaqMan RT-PCR master mix, 0.4 µM of forward primer, 0.6 µM of reverse primer, 0.25 µM of probe, a 0.125 ng/µL RNA sample, and 10 µL of ultra-pure water q.s. The primers and probes used are listed in Table 1. All reactions were run in a QuantStudio 6 Flex (Applied Biosystems^®^, Waltham, MA, USA) with the following PCR conditions: reverse transcription at 50 °C for 5 min and qPCR conditions at 95 °C for 20 s, followed by 40 amplification cycles of 95 °C for 15 s and 60 °C for 1 min [44].

The infection rate (IR) was calculated by dividing the number of mosquitoes that presented YFV RNA in both the abdomen and head by the total number of tested mosquitoes multiplied by 100. The dissemination rate (DR) was calculated by dividing the number of mosquitoes with YFV RNA in their heads by the total number of tested mosquitoes multiplied by 100. The midgut infection rate (MIR) was calculated by dividing the number of mosquitoes with YFV RNA only in the abdomen (and absent in the heads) by the total number of tested mosquitoes, multiplied by 100. Viral load was calculated using a standard curve generated by serial ten-fold dilution (ranging from 75 to 7.5 × 10^6^ PFU) of YFV RNA extracted from a culture medium containing the virus transcript.

### 2.6. DNA Extraction and kdr Genotyping

Thorax DNA was extracted, and an SNP *kdr* genotyping approach was employed to evaluate the mutation at the *Na_V_* site Phe1534Cys [46,47]. Briefly, reactions consisted of 1× TaqMan Genotyping Master Mix (ThermoFisher ^®^), 1× of the respective Custom TaqMan SNP Genotyping Assay (Table 2), 20 ng of DNA, and 10 μL of ultra-pure water q.s. These reactions were run on a QuantStudio 6 Flex (Applied Biosystems^®^, Waltham, MA, USA) under standard conditions. PCR conditions included 45 cycles of DNA denaturation (95 °C for 15 s) and annealing of primers and probe (60 °C for 1 min), followed by a final extension (60 °C for 30 s). Negative and positive controls (for each 1534 genotype) were used for the reaction. Genotyping analysis was performed using the online software Genotype Analysis Module V3.9 (Applied Biosystems, on the ThermoFisher Cloud platform) using the profile presented in the CT 40.

### 2.7. HRM Analysis and DNA Sequencing

In high-resolution melting (HRM) analysis, DNA molecules are replicated similarly to a standard qPCR reaction, but with the addition of a dye that emits fluorescence when it intercalates into the double strand of DNA. This results in the formation of fluorescence decay curves, reflecting dye dissociation from the DNA double strand at different temperatures, allowing for qPCR detection of different patterns corresponding to various SNPs and differentiating the mutations in the *Na_V_
* SNPs Phe1534Cys and Thr1520Ile. HRM analysis was performed as described elsewhere [32]. Briefly, reactions were conducted using 5 µL of the MeltDoctor HRM Master Mix kit (ThermoFisher^®^ Waltham, MA, USA), 0.3 µM of each primer, 1.0 µL of DNA (at 20 ng/µL), and 10 μL of ultra-pure water q.s. Reactions were carried out in a QuantStudio 6 real-time thermocycler (ThermoFisher^®^) under standard conditions: 95 °C for 10 min, followed by 40 cycles at 95 °C for 15 s and 60 °C for 1 min. After amplification, the HRM stage was initiated, gradually increasing the temperature from 60 °C to 95 °C to denature the double strand and generate fluorescent decay curves. Nucleotide sequence patterns obtained from the HRM analysis were confirmed by DNA sequencing. For sequencing, PCR was performed with 1× Phusion buffer, 3% DMSO, 0.5 µM of the primers (Table 3), and 25 µL of ultra-pure water q.s. Thermocycling conditions included an initial polymerase activation step at 98 °C for 30 s, followed by 35 cycles at 98 °C for 10 s, annealing at 61 °C for 15 s, extension at 72 °C for 30 s, and a final extension step at 72 °C for 7 min. Amplified DNA was purified using the AnyPrepMag PCR Clean-Up Kit (Axygen^®^, Union City, CA, USA) according to the manufacturer’s instructions. For sequencing reactions, the BigDye^®^ Terminator v3.1 Cycle Sequencing kit (ThermoFisher^®^) was used with 1 µL of amplified and purified DNA and 1 µM of each primer. The reactions were subjected to an initial step of 1 min at 96 °C, followed by 60 cycles at 96 °C for 10 s, 50 °C for 5 s, and 60 °C for 2 min. Sequencing of both strands of the amplified products was performed using the FIOCRUZ DNA-sequencing platform.

### 2.8. Statistical Analysis

Comparisons of the infection rate (IR), dissemination rate (DR), and midgut infection rate (MIR) were performed using the chi-square test followed by Fisher’s exact test (*p*< 0.05). Differences in viral load were evaluated using the non-parametric Mann–Whitney U test (*p* < 0.05). All plots and statistical analyses were conducted using GraphPad Prism^®^ 5 software.

## 3. Results

### 3.1. Susceptibility of Ae. aegypti from Pakistan to YFV

#### 3.1.1. YFV Infection Rates in Mosquitoes

To assess the susceptibility of *Ae*. *aegypti* from Pakistan to YFV, we measured the viral infection rates following an infectious blood meal containing two YFV strains (Figure 1 and Table 4). Pakistani *Ae*. *aegypti* were orally susceptible to YFV, with infection rates for both YFV strains comparable to those of Rockefeller mosquitoes (YFV PR4408/2008: Pakistani *Ae*. *aegypti*, IR = 83.7%; Rockefeller, IR = 92.7%; *p* = 0.6643; YFV ES504: Pakistani *Ae*. *aegypti*-IR = 61.7%, Rockefeller-IR = 41.0%, *p* = 0.2323, Figure 1A,B and Table 4). Evaluation of dissemination rates (DR) of *Ae*. *aegypti* from Pakistan also did not show significant differences compared with Rockefeller (YFV PR4408/2008: Pakistani *Ae*. *aegypti*, DR = 53.5%; Rockefeller, DR = 85.7%; *p* = 0.0556; YFV ES504, Pakistani *Ae*. *aegypti*, DR = 54.8%, Rockefeller, DR = 28.4%, *p* = 0.1320, Figure 1C,D and Table 4). Additionally, we observed midgut infection rates (MIR), where YFV was detected in the mosquito abdomen without dissemination to the head, for both YFV strains. However, there were no significant differences between *Ae. aegypti* from Pakistan and Rockefeller (YFV PR4408/2008: Pakistani *Ae*. *aegypti*, MIR = 30.2%; Rockefeller, MIR = 7.1%, *p* = 0.08; YFV ES504-Pakistani *Ae*. *aegypti*, MIR = 6.7%; Rockefeller, MIR = 12.5%; *p* = 0.6073, Figure 1E,F and Table 4).

Our findings on YFV IR and DR were compared with those of previous records for Asian *Ae. aegypti* populations (Figure 2). Despite the significant variation in the YFV titers of the blood meals offered to mosquitoes across these studies, which precludes a direct quantitative comparison of IR and DR, our results align with findings from at least one study involving Asian *Ae. aegypti* populations [16].

#### 3.1.2. YFV Viral Load

To compare the infection burden of *Ae*. *aegypti* from Pakistan and the Rockefeller lineage, we assessed the YFV viral load in the mosquito heads and abdomens (Figure 3). The mean viral loads in the head were similar between the mosquito populations (YFV PR4408/2008: Pakistani *Ae*. *aegypti*-Mean = 2.6 × 10^6^ ± 9.4 × 10^5^, Rockefeller-Mean = 1.1 × 10^6^ ± 2.5 × 10^5^, *p* = 0.2499; YFV ES504: Pakistani *Ae*. *aegypti*-Mean = 1.2 × 10^6^ ± 3.7 × 10^5^, Rockefeller-Mean = 0.9 × 10^6^ ± 3.7 × 10^5^, *p* = 0.7014, Figure 3A,B). For the PR4408/2008 YFV strain, the mean viral load differences in the abdomens could not be calculated because only one Rockefeller mosquito was infected in the abdomen, precluding comparison for this virus strain (Figure 3C). However, for the ES504 YFV strain, no significant differences were observed (Pakistani *Ae*. *aegypti*: Mean = 0.1 × 10^5^ ± 3.74 × 10^3^, Rockefeller: Mean = 2.4 × 10^5^ ± 1.2 × 10^5^, *p* = 0.1960, Figure 3D).

### 3.2. Ae. aegypti from Pakistan: kdr Genotypes × YFV Infection

We genotyped two SNPs in the *Na_V_* IIIS6 segment from the 72 Pakistani *Ae. aegypti* mosquitoes that were orally infected with YFV using HRM analyses and TaqMan probes T1520I and F1534C (Figure 4). HRM analyses discriminated two genotypes out of the nine possibilities (1520 + 1534: TTFF, TTFC, TTCC, TIFF, TIFC, TICC, IIFF, IIFC, and IICC). Together with TaqMan genotyping for the F1534C SNP, we identified two predominant *kdr* genotypes, double homozygote *kdr* IICC (35%) and heterozygote TIFC (65%).

We amplified and sequenced a 354 bp DNA fragment corresponding to the IIIS6 *Na_V_* segment from four IICC and four TIFC samples, confirming their genotypes in all cases (Figure 5). Sequences have been deposited in GenBank under accession number MN602779.

There was no significant difference in DR between IICC and TIFC mosquitoes for either YFV strain (YFV PR4408/2008: IICC-DR = 66.7%, TIFC-DR = 47.8%, *p* = 0.2568; YFV ES504: IICC-DR = 50%, TIFC-DR = 63.9%, *p* = 0.5607, Figure 6A,B and Table 5). Similarly, MIR did not differ significantly between IICC and TIFC mosquitoes for the YFV PR4408/2008 strain (IICC-MIR = 20%, TIFC-MIR = 33.3%, *p* = 0.3786; Figure 6C and Table 5). For mosquitoes fed with the YFV ES504 strain, only two infected abdomens were observed, both displaying the TIFC genotype.

Viral titers were not significantly different between homozygote (IICC) and heterozygote (TIFC) *kdr* mosquitoes in both the head and abdomen samples (Figure 7). For the heads, mean viral loads were comparable between IICC and TIFC mosquitoes infected with YFV PR4408/2008 (IICC-Mean = 4.14 × 10^6^ ± 1.86 × 10^6^, TIFC-Mean = 1.54 × 10^6^ ± 9.2 × 10^5^, *p* = 0.1489, Figure 7A) and YFV ES504 (IICC-Mean = 9.94 × 10^5^ ± 9.03 × 10^5^, TIFC-Mean = 1.27 × 10^6^ ± 4.17 × 10^5^, *p* = 0.2345, Figure 7B). Similarly, in the abdominal samples, there were no significant differences in viral loads between IICC and TIFC mosquitoes infected with YFV PR4408/2008 (IICC-Mean = 9.05 × 10^0^ ± 2.18 × 10^4^, TIFC-Mean = 1.67 × 10^5^ ± 4.8 × 10^4^, *p* = 0.3152, Figure 7C).

## 4. Discussion

This study assessed the susceptibility of an *Ae. aegypti* population from Pakistan to the yellow fever virus (YFV), shedding light on potential factors contributing to the absence of YF in Asia. Our findings indicate that the Pakistani *Ae. aegypti* mosquitoes are orally susceptible to YFV infection, regardless of their *kdr* genotypes, and exhibit infection rates (IR) and dissemination rates (DR) comparable to those of the Rockefeller strain, a well-characterized reference in vector competence studies.

Few studies have evaluated susceptibility of Asian *Ae*. *aegypti* populations to YFV, underscoring the novelty of our findings. Notably, a previous study on Pakistani *Ae. aegypti* mosquitoes reported lower DR (30.1%) compared to our results (53.5–54.8%), albeit under different experimental conditions that may have influenced these discrepancies [26]. In that study, the virus load titer was not described, and the mosquitoes belonged to a colony that was maintained under laboratory conditions for many generations. Prolonged laboratory colonization may alter mosquito physiology and immune responses to viral infections [49,50,51,52]. Interestingly, our study aligns with findings from other Asian populations, including Singapore, Vietnam, and Laos, where *Ae. aegypti* is highly susceptible to YFV infection [16,19]. The similar IR, DR, and viral loads observed between Pakistani and Rockefeller mosquitoes with both viral strains tested suggest that *Ae*. *aegypti* across different geographic regions maintained robust vector competence for YFV, implying that vector competence alone may not limit the emergence of YF in Asia.

However, the dissemination rate (DR) of *Ae*. *aegypti* from Pakistan infected with the YFV PR4408/2008 strain showed a marginally lower tendency than Rockefeller (*p* = 0.0556). Vector competence is a multifaceted trait influenced by various factors, including the molecular mechanisms of innate immunity (involving pattern recognition pathways and small RNAs), physical barriers (such as peritrophic matrix and hemocele), physiological barriers (including low midgut pH and barriers in the midgut and salivary gland), small interfering RNA (siRNA) response, and microbiota interactions [10,53,54,55]. These factors can act independently or synergistically to reduce viral multiplication and dissemination to other tissues, potentially impeding the vector transmission of the arbovirus (refractoriness). In our study, a lower DR tendency was observed in the Pakistani *Ae*. *aegypti*, which may suggest the presence of a physical barrier such as the midgut escape barrier (MEB), which has been reported to inhibit viral dissemination in other contexts [56,57,58]. Further investigation is warranted to confirm the presence and mechanism of MEB in Pakistani *Ae. aegypti*.

A mathematical model has suggested that the absence of urban yellow fever in Asia cannot be sufficiently explained by mosquito vector competence to YFV alone. This suggests that other factors, such as impaired YF transmission in vectors co-infected with dengue and host cross-immunity to YF following prior flavivirus infections, likely play significant roles [59]. Some studies have explored this hypothesis, showing that the presence of dengue or Zika virus-specific immunoglobulin antibodies significantly reduces YFV viremia and disease severity [60,61]. However, comprehensive epidemiological and serological studies are needed in Asia to clarify the role of cross-reactivity in providing protection against YFV infection. Establishing an arboviral transmission cycle in a region involves complex interactions between viral and vector genetics, host density, and susceptibility factors [62]. Some studies have demonstrated that mutations in arboviruses can enhance replication and transmission, directly influencing the incidence of environmental viral cycles [63,64,65]. For example, mutations in dengue (DENV) and YFV envelope proteins have been associated with increased virulence and infectivity in hosts [66,67], while a specific mutation in the envelope protein gene (E1-A226V) of chikungunya virus (CHIKV) has increased its infectivity in *Aedes albopictus* [68]. Similarly, genetic alterations in YFV or its sylvatic hosts could potentially facilitate the establishment of transmission cycles in Asia. However, a sustained sylvatic YF transmission cycle among monkey populations would require certain demographic and susceptibility factors, such as a high annual birth rate (e.g., 400 per 1000), to prevent herd immunity from suppressing the arboviral cycle [69]. Asian native non-human primates (NHPs) have shown high susceptibility to YFV infections, indicating that ecological and serological surveys are needed to assess the potential of Asian NHPs to sustain a YFV sylvatic cycle [70,71]. Thus, the absence of the YF in Asia likely involves a combination of these factors.

Vector control strategies based on chemical use exert strong selection pressure on natural populations, leading to physiologically significant changes within a few generations. These changes may disrupt genomic balance, potentially posing a disadvantage to insects in environments without insecticide exposure. *kdr* mutations are among the primary mechanisms conferring resistance to pyrethroid insecticides [72]. In a previous study, we identified the presence of *kdr* mutations T1520I and F1534C in *Ae. aegypti* larvae in Pakistan [32]. As *kdr* mutations generally lead to a loss of fitness, it is hypothesized that the vectorial capacity of insecticide-resistant mosquitoes may be affected [73,74,75]. The influence of insecticide resistance mechanisms, such as *kdr*, on pathogen infection has been addressed in some mosquito species [74,76,77]. For instance, *Anopheles* mosquitoes with L1014F or L1014S *kdr* mutations showed a higher prevalence of *Plasmodium falciparum* infection but a lower parasite burden (in L1014F mosquitoes), suggesting that the pleiotropic effects associated with *kdr* mutations may influence the intricate infection process [38,40,78]. In *Ae*. *aegypti* mosquitoes with V1016I and F1534C *kdr* mutations, higher Zika infection rates (IR) and dissemination rates (DR) were observed at different time points post-infection [79]. More recently, we observed that in natural *Ae. aegypti* population from Brazil mosquitoes were infected with chikungunya virus (CHKV) regardless of their *kdr* genotype [80], which is in agreement with our study, as we did not observe a correlation between the mosquito *kdr* genotypes and YFV infection. Further evaluations and comparisons with diverse natural populations and *kdr* homozygous laboratory strains are necessary to better understand the influence of *kdr* mutations on YFV and other arbovirus infections in *Ae*. *aegypti*.

Overall, the results presented herein indicate that *Ae. aegypti* from Pakistan is susceptible to YFV infection, regardless of the *kdr* genotype, indicating that many other factors might influence the absence of yellow fever disease in this country.

## Figures and Tables

**Figure 1 insects-16-00033-f001:**
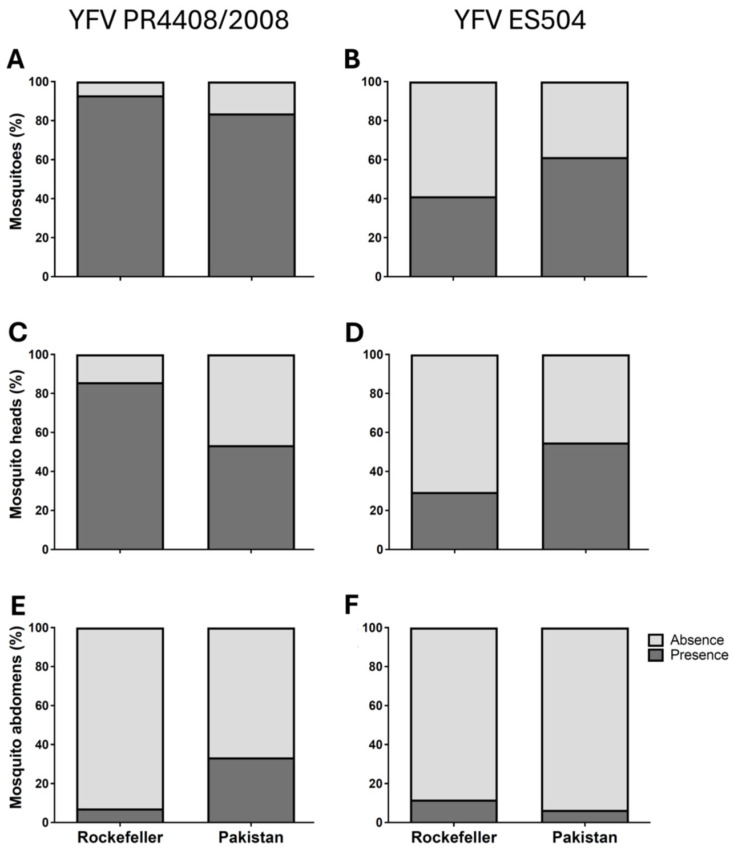
Infection of Pakistani *Ae*. *aegypti* 14 days after challenge with one of two YFV strains. Graphs (**A**,**C**,**E**): infection with YFV PR4408/2008; Graphs (**B**,**D**,**F**): infection with YFV ES504; Graphs (**A**,**B**): infection rate (IR); Graphs (**C**,**D**): dissemination rate (DR); Graphs (**E**,**F**): midgut infection rate (MIR).

**Figure 2 insects-16-00033-f002:**
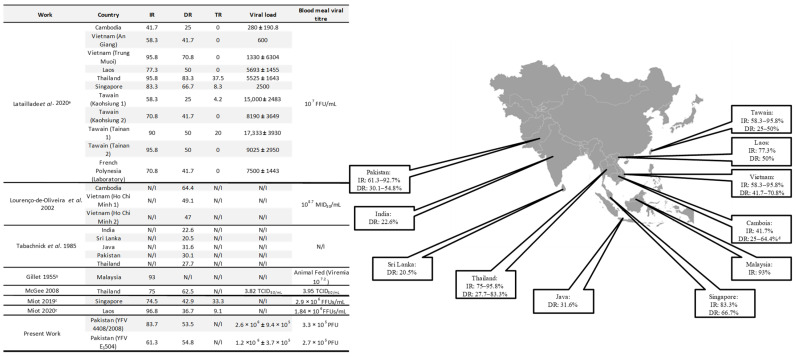
YFV infection rate (IR), dissemination rate (DR), and viral load of *Ae*. *aegypti* mosquitoes in other Asian countries. ^a^ Viral load—head titer from day 21 post blood meal (pbm); ^b^ IR from 30 days pbm; ^c^
*Ae*. *malayensis*; ^d^ A higher dissemination rate (DR) compared to the infection rate (IR) was achieved, as indicated by the elevated dissemination value reported in a previous study. Lataillade et al. 2020 [16]; Lourenço-de-Oliveira et al. 2002 [19]; Tabachnick et al. 1985 [26]; Gillet 1955 [25]; McGee 2008 [27]; Miot 2019 [28]; Miot 2020 [48].

**Figure 3 insects-16-00033-f003:**
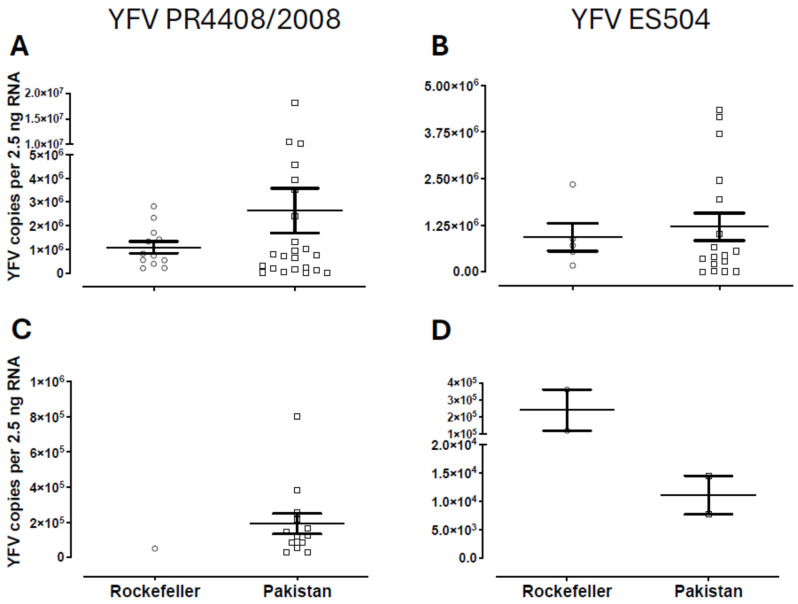
Head and abdominal viral loads in *Ae*. *aegypti* 14 days after challenge with YFV. Graphs (**A**,**C**): infection with YFV PR4408/2008; Graphs (**B**,**D**): infection with YFV ES504; Graph (**A**,**B**): heads; Graph (**C**,**D**): abdomens; each dot represents a single mosquito; error bar: standard error of the mean (SEM).

**Figure 4 insects-16-00033-f004:**
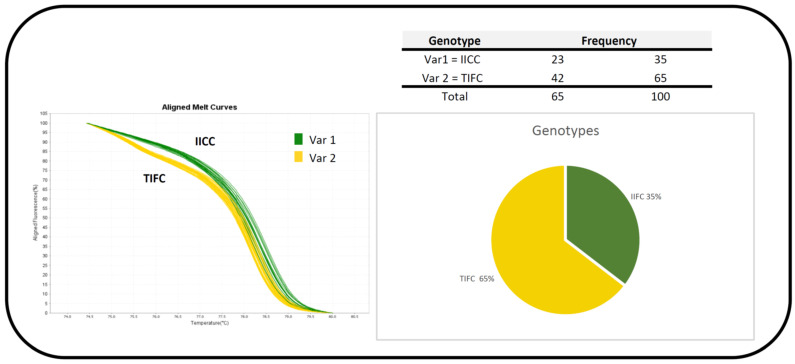
HRM curves and *kdr* frequencies from the IIIS6 segment in *Na_V_* region of Pakistani *Ae*. *aegypti*. Var: variant; II: isoleucine (Ile) homozygosity; TI: threonine (Thr)/isoleucine heterozygosity; CC: cysteine (Cys) homozygosity; F/C: phenylalanine (Phe)/cysteine heterozygosity.

**Figure 5 insects-16-00033-f005:**
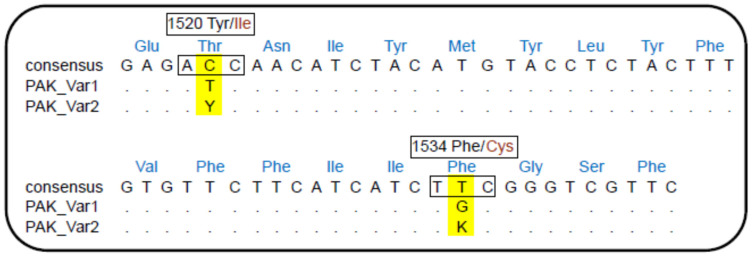
DNA sequence alignment of *Na_V_* IIIS6 region of the two variants found in the HRM reaction. Y: C/T heterozygote; K: T/G heterozygote. The 1520 and 1534 codons are depicted, with the mutant amino acid shown in red, while the nucleotide variants in these codons are highlighted.

**Figure 6 insects-16-00033-f006:**
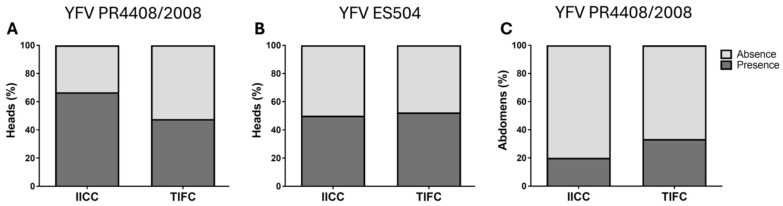
Effect of *kdr* 1520I + F1534C mutations on YFV infection in Pakistani *Ae*. *aegypti*. YFV PR4408/2008 infection: graphs (**A**,**C**); YFV ES504 infection: graph (**B**); Graphs (**A**,**B**): dissemination rate (DR); Graph (**C**): midgut infection rate (MIR).

**Figure 7 insects-16-00033-f007:**
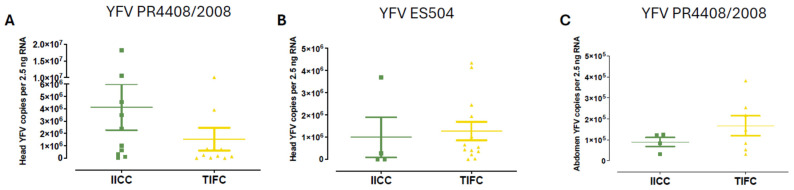
YFV viral load in the head and abdomen of *Ae*. *aegypti* carrying the *kdr* 1520I + F1534C mutation. Graphs (**A**,**C**): YFV PR4408/2008 infections. Graph (**B**): YFV ES504 infection. Graphs (**A**,**B**): mosquito heads. Graph (**C**): mosquito abdomen. Each dot represents a mosquito. Error bars: standard error of mean (SEM).

**Table 1 insects-16-00033-t001:** Primers and probes for RT-qPCR YFV detection.

Primer or Probe	Sequence	Position *
Primers		
YF-all-forward	5′-GCTAATTGAGGTGYATTGGTCTGC-3′	15–38
YF-all-reverse	5′-CTGCTAATCGCTCAAMGAACG-3′	83–103
Probes		
YF-all	5′-FAM-ATCGAGTTGCTAGGCAATAAACAC-TMR-3′	

TMR, 6-carboxytetramethylrhodamine; FAM, 6-carboxyfluorescein. * Positions are indicated in relation to the GenBank sequence AY640589.1 for the yellow fever virus Asibi strain. The primer and probe sequences were obtained from Domingo et al. 2012 [45].

**Table 2 insects-16-00033-t002:** Primers and probes for TaqMan SNP genotyping qPCR assays of *kdr* mutations in *Ae*. *aegypti*.

*Na_V_* Site, Segment Primers and Probe	Assay ID *	Variation	Sequence
Primer			
1534, IIIS6	AHWSL61	TTC/TGC (Phe/Cys)	for: 5′-TCGCGAGACCAACATCTACATG-3′ rev: 5′-GATGATGACACCGATGAACAGATTC-3′
Probe	-	-	Phe: 5′-VIC-AAC GAC CCG AAG ATGANFQ-3′ Cys: 5′-FAM-ACG ACC CGA CGA TGA-NFQ-3′

* Identification using the customized TaqMan SNP Genotyping Assay (ThermoFisher^®^).

**Table 3 insects-16-00033-t003:** Sequences of primers used for amplification of the IIIS6 region of *Na_V_*.

**HRM**	**Sequence**
*For 9*	5′-TGGGAAAGCAGCCGATTCG-3′
*Rev 8*	5′-GAACAGATTCAGCGTGAAGAACG-3′
**Sequencing (IIIS6)**	**Sequence**
*For: 31P*	5′-TCGCGGGAGGTAAGTTATTG-3′
*Rev: 31Q*	5′-GTTGATGTGCGATGGAAATG-3′

**Table 4 insects-16-00033-t004:** The infection rate, dissemination rate, and midgut infection rate of *Ae*. *aegypti* strain orally fed one of the two YFV strains.

YFV Strain	*Ae*. *aegypti* Strain	IR	*p* Value *	DR	*p* Value	MIR	*p* Value *
PR4408/2008	Rockefeller	92.7 ± 7.3%	0.6643	85.7 ± 1.1%	0.0556	7.1 ± 7.1%	0.08
	Pakistan	83.7 ± 2.7%		53.5 ± 5.6%		30.2 ± 3.0%	
ES504	Rockefeller	61.7 ± 11.7%	0.2323	28.5 ± 16.0%	0.132	12.5 ± 12.5%	0.6073
	Pakistan	41.0 ± 3.5%		54.8 ± 12.0%		6.7 ± 0.4%	

IR: infection rate; DR: dissemination rate; MIR: midgut infection rate; IR, DR, and MIR values: mean ± SEM; * chi-square test followed by Fisher’s exact test (*p* < 0.05).

**Table 5 insects-16-00033-t005:** Dissemination rates and midgut infection rates of *Ae*. *aegypti* from Pakistan with different *kdr* genotypes orally fed one of the two YFV strains.

YFV Strain	*Ae. aegypti kdr* Genotype	DR	*p* Value *	MIR	*p* Value *
PR4408/2008	TIFC	47.8 ± 7.8%	0.2568	33.3 ± 4.2%	0.3786
	IICC	66.7 ± 0.02%		20.0 ± 5.3%	
ES504	TIFC	63.9 ± 13.9%	0.5607	-	-
	IICC	50.0 ± 16.7%		-	

DR: dissemination rate; MIR: midgut infection rate; DR and MIR values: mean ± SEM; * chi-square test followed by Fisher’s exact test (*p* < 0.05).

## Data Availability

All data generated or analyzed during this study are included in this published article.

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
