# Peer review of "Evaluation of Yellow Fever Virus Infection in Aedes aegypti Mosquitoes from Pakistan with Distinct Knockdown Resistance Genotypes"

_insects, 2024, doi:10.3390/insects16010033_

Round 1
Reviewer 1 Report
Comments and Suggestions for Authors
This is a tidy manuscript detailing the curious lack of yellow fever virus in Asia by looking at Pakistani Aedes aegypti. The authors do a nice job of explaining the need to explore this topic, provide rationale for exploring the kdr genotypes, and demonstrate that Pakistani Ae. aegypti are susceptible to acquiring YFV. All this evidence points to some, yet unknown, cause for the absence of yellow fever from the region.
Items to improve this work: I think the authors did a good job on this manuscript. My greatest criticism is for the visual presentation of the data. In Figures 1 and 6, I would like to see the error associated with the data represented visually. While it is clearly indicated in the text that these are not statistically different values, a set of error bars would allow the reader to make that conclusion on their own. Also placing the YFV stain above the column of graphs would allow the reader to easily distinguish between the data sets (left & right)
Figure 2 felt odd in the results section considering that it compares this work with previous work. This is more suited for the discussion section, since it does not describe the findings but rather puts them in context.
Figure 3 would also be improved by placing column heading for the A&C, and B&D. In addition to side labels for Head (A&B) and Abdomen/midgut (C&D).
Figure 4 - I would recommend removing the “Genotypes 1520 + 1534” as the rest of the terminology is Var1 and Var2. It is distracting at minimum, if not confusing.
Figure 5 - I do not care for the use of “Y” for pyrimidines and “K” for ketones. In a figure where you are using 1-letter notation for the amino acids in the first row you should not use degenerate nucleic acid codes outside ATGC. You should use the three letter amino acid notation in the first row or use C/T and G/T in the variants.
Figure 6 & 7 - use the column and row heading as described above and don’t center panel C.
Very minor points:
Line 19 - Yellow Fever Virus is spelled “Yello fever virus”
Line 19-20 - you say “standard mosquito strain” and I would say “laboratory strain”
Line 48 - “death rate raging” should be death rate ranging.
Line 141 - should not assume B.O.D. is commonplace, please do not abbreviate
Line 192 - should not assume HRM is commonplace, please use the full text before abbreviation.
Line 304 - says Viral titter, should say Viral titer
Author Response
Reviewer 1
This is a tidy manuscript detailing the curious lack of yellow fever virus in Asia by looking at Pakistani Aedes aegypti. The authors do a nice job of explaining the need to explore this topic, provide rationale for exploring the kdr genotypes, and demonstrate that Pakistani Ae. aegypti are susceptible to acquiring YFV. All this evidence points to some, yet unknown, cause for the absence of yellow fever from the region.
Items to improve this work: I think the authors did a good job on this manuscript. My greatest criticism is for the visual presentation of the data. In Figures 1 and 6, I would like to see the error associated with the data represented visually. While it is clearly indicated in the text that these are not statistically different values, a set of error bars would allow the reader to make that conclusion on their own. Also placing the YFV stain above the column of graphs would allow the reader to easily distinguish between the data sets (left & right)
>>We thank Reviewer 1 for their appreciation and encouragement. Regarding Figures 1 and 6, it was not possible to add error barsin these types of plots: in Figure 1, we analyzed the absence/presence of the virus in Rock and Pakistani mosquito groups as a percentage of the total in each group,totaling 100%. Similarly, in Figure 6, we compared different kdr genotype groups, based on their respective totals. To clarify the absence of significant differences, we added two tables (Table 4 e Table 5), with numerical values and statistical analyses to complement these comparisons. These new tables were opportunely addressed in the text (lines 249, 304).
Figure 2 felt odd in the results section considering that it compares this work with previous work. This is more suited for the discussion section, since it does not describe the findings but rather puts them in context.
>> We agree that this Figure does not show any findings from our experiments; however, we believe it effectively illustrates previous results in a schematic form. Since space is typically limited for figures in the Discussion section, we have chosen to retain it in the Results section.
Figure 3 would also be improved by placing column heading for the A&C, and B&D. In addition to side labels for Head (A&B) and Abdomen/midgut (C&D).
>>Figure improved.
Figure 4 - I would recommend removing the “Genotypes 1520 + 1534” as the rest of the terminology is Var1 and Var2. It is distracting at minimum, if not confusing.
>>Figure improved.
Figure 5 - I do not care for the use of “Y” for pyrimidines and “K” for ketones. In a figure where you are using 1-letter notation for the amino acids in the first row you should not use degenerate nucleic acid codes outside ATGC. You should use the three letter amino acid notation in the first row or use C/T and G/T in the variants.
>> Point noted. We used the Y (for T or C) and K (for T or G) to represent nucleotide variations called by Var2 in the HRM analyses. We have now updated the aminoacids to the 3-letter notation in the first row.
Figure 6 & 7 - use the column and row heading as described above and don’t center panel C.
>>These Figures were improved as recommended.
Very minor points:
Line 19 - Yellow Fever Virus is spelled “Yello fever virus”Done
Line 19-20 - you say “standard mosquito strain” and I would say “laboratory strain”Done
Line 48 - “death rate raging” should be death rate ranging.Done
Line 141 - should not assume B.O.D. is commonplace, please do not abbreviateDone
Line 192 - should not assume HRM is commonplace, please use the full text before abbreviation.Done
Line 304 - says Viral titter, should say Viral titerDone

Reviewer 2 Report
Comments and Suggestions for Authors
The authors investigated the susceptibility of Aedes aegypti mosquitoes from Pakistan to Yellow Fever Virus (YFV) infection and the potential influence of kdr genotypes, which provides valuable insights into understanding why yellow fever has not emerged in Asia to date.
The main strengths of this study are:
1. The experimental design is well-conceived, employing two different YFV strains for infection experiments and comparing the results with standard reference strains.
2. The methodology is scientifically rigorous, incorporating essential procedures such as viral infection, RNA extraction, and genotyping, all conducted according to established scientific protocols.
3. The results demonstrate that Pakistani Ae. aegypti are indeed susceptible to YFV infection, with infection rates comparable to control groups, providing crucial experimental evidence for understanding the absence of yellow fever outbreaks in Asian regions.
However, several aspects of the manuscript require improvement:
1. The manuscript lacks critical raw data. I recommend including comprehensive experimental data in the supplementary materials. The provision of these raw data would enable other researchers to validate the findings and conduct further analyses.
2. The presentation of figures and tables needs refinement. The current format, with figures enclosed in rounded rectangles, compromises both readability and professional appearance. I suggest redesigning the graphical presentations to enhance clarity and adopting standard scientific journal formatting conventions. Additionally, figure legends and labels should be uniformly adjusted for consistency and clarity.
3. More detailed statistical analyses are recommended, particularly regarding the relationship between different kdr genotypes and viral infection rates. This would strengthen the conclusions of the study.
In conclusion, this study provides valuable experimental data and scientific insights that enhance our understanding of Ae. aegypti susceptibility to YFV, particularly in the Asian context. I recommend resubmission after incorporating complete raw data and improving the quality of graphical presentations. With these modifications, the manuscript should meet publication standards.
Author Response
Reviewer 2
The manuscript tries to shed light on the absence of yellow fever in Asia, despite the presence in the region of the competent vector Aedes aegypti. The study demonstrates that Pakistani Ae. aegypti mosquitoes can be infected with no significant differences compared to a reference American strain. Furthermore, the connection between kdr mutations, causing resistance to pyrethroids, and capability to be infected with yellow fever virus, is investigated, resulting in absence of correlation, differently from previous studies on different mosquitoes and viruses.
The manuscript is well written, and the experiments are reported in a detailed way. However, I noticed a general lack of analysis depth on the results obtained and on the rationale behind the study. It would be wise to highlight why it is important to screen the susceptibility to yellow fever virus infection in mosquitoes with different kdr genotypes, reporting carefully previous cases of increased susceptibility to viruses because of kdr resistance; the finding, in this study, of absence of correlation between kdr and YFV susceptibility is opposite to previous evidence, hence it could be worth trying to understand better the possible reasons of this unexpected result. The study may also look incomplete because of the many questions left open about the reasons of absence of YFV in Asia, which are just described with no proper in-depth analysis. Further considerations in the Discussion could give the manuscript more completeness. I recommend this manuscript for publishing in Insects after revisions.
>>We thank “Reviewer 2” for their meticulous review and encouraging feedback. We have refined the text to better highlight the rationale behind our hypothesis: that an Asian Ae. aegypti population can be efficiently infected with distinct YFV strains, regardless of its pyrethroid resistance genotype (in this case, associated with kdr mutations), which are often suggested to strongly influence various physiological traits in the organism. Given our findings of no significant differences, we emphasize that factors beyond the mosquito genetic background and virus strains may be contributing to the absence of yellow fever transmission in Asia.
General comments:
- The logic behind the abstract is fragmented, the connection between kdr and YFV infection should be stated also there, and not only in Introduction and Discussion.
>> Agreed. We have revised the abstract to address this comment.
- At the end of the Introduction, the hypothesis of kdr mutations increasing susceptibility to YFV is introduced in a very concise way which could be explained better (also considering that this hypothesis will be refused in the manuscript).
>>This paragraph has been revised to clarify our hypothesis.
- Most of the figure captions could be rephrased to be easier to read, since the high schematic way they are written can be confusing.
>>We revised all the figures, including the “Reviewer 1” concerns.
- The Discussion contains a lot of information but lacks analysis depth. Many possible reasons for absence of yellow fever in Asia are given, giving the impression of a list without critical analysis of what could be causing the difference between Asian environment and African and American one.
>> We revised the Discussion section considering this advice.
- The same can be said about the discussion on kdr and susceptibility to YFV, but one important thing is that the mentioned references mostly suggest increased/altered susceptibility to viruses because of kdr mutation, while the previous publication from your group (75) stated that, in Brazilian Ae. aegypti mosquitoes, kdr did not cause any differences in infection with chikungunya virus (hence agreeing with the present manuscript).
>>Yes, we agree. In the revised text we highlighted these findings.
- Particular attention should be given to standardising punctuation and structure of phrases; some words and phrasings may be changed.
>> We appreciate this reminder. We also noticed that our original text was edited to fit the journal’s format, which altered some of our sentences. We have corrected these in addition to addressing the reviewers’ comments.
Detailed comments:
- Line 26-27: the link between yellow fever virus and mutations causing insecticide resistance is not explicated here.
Corrected
- Line 30: as it is the first mention of YFV, it should be explained: “genetic differences in yellow fever virus (YFV) strains”.Done
- Line 33: I would add “female adults”.Done
- Line 53: “sylvatic vectors, such as…”Done
- Line 53-55: I would rephrase: “the sylvatic cycle, in which the circulation between sylvatic vectors, such as Haemagogus and Sabethes genera, and non-human primates (NHPs) can accidentally infect humans; …”Done
- Line 56: A comma could be needed: “and the urban cycle, in which humans…”Done
- Line 61: Maybe: “and all three transmission cycles”.Done
- Line 61: Rephrase: “with the intermediate one usually serving as a bridge for the urban one”.Done
- Line 66: “involving immune responses, microbiota and genetic factors, …”Done
- Line 67: “… and viral genotypes, important determinants…”Done
- Line 68-69: I would rephrase to give more continuity to the text: “Indeed, one extrinsic factor with a key role for human YFV infections is the presence and distribution of susceptible mosquito vectors, such as Aedes …”Done
- Line 75-77: Rephrase needed: “… in Asian countries, while other Ae. aegypti arboviruses, such as dengue and chikungunya, are responsible for epidemies in the Asia-Pacific region, as well as in the American and African continents”Done.
- Line 81-84: I would give a different tone: “Although natural populations (…) and transmission of the virus, laboratory infection assays demonstrated that…”Done
- Line 91: This part needs to be explained better; it is too stretched.Done
- Line 95-97: “… for mutations in several Nav sites, both worldwide-spread, as the F1534C substitution, ad regional-specific, as the V410L and V1016I mutations in the Americas and in Africa and the S989P, V1016G and T1520I mutations in Asia”.Done
- Line 97: I would add: “In Ae. aegypti populations from Pakistan”Done
- Line 98: I would add: "… forming two different kdr haplotypes:”Done
- Line 100: This is a crucial point to explain the rationale of the research and I think it should be described in a more detailed way. The idea behind is: since the loss in fitness and the reduction in vector capacity have already been studied for several mosquitoes and viruses, but no studies have been reported for Ae. aegypti infections with YFV, it would be important to check if kdr mosquitoes have enhanced susceptibility to this virus. This concept should appear clearer, without jumping to the Discussion or reading the mentioned references.Thank you.We expanded this idea in the paragraph and added the corresponding references
- Line 109: “… as colonies in insectaries”.Done
- Line 115: in “2L”, it would be better to separate the number from the measure unit, since in the next lines they are always separated.Done
- Line 129: better to add a comma before “isolated”.Done
- Line 134: “24-hours”.Done
- Line 141: need to explain “B.O.D.”.Done
- Line 146: “Mosquitoes”.Done
- Line 153: “… as described by the…”Done
- Line 156: “… instructions; the nucleic acid was stored…”Done
- Line 156: I would speak about “reactions” as plural.Done
- Line 174-175: “Positions are indicated in relation to...”Done
- Line 189: “Ae. aegypti”.Done
- Line 196: I would say: “… temperatures, allowing for qPCR detection…”Done
- Line 205: Why are the degrees symbols here looking different compared to other parts of the manuscript?It was a different pattern. We corrected this issue
- Line 214: I would use “used” instead of “utilized”.Done
- Line 247-248: “Graph” should be written as plurals.Done
- Line 249-250: I would rephrase: “Our findings of YFV IR and DR were compared with previous records for Asian Ae. aegypti populations (Figure 2)”.Done
- Line 270-273: I would suggest keeping using the same punctuation: “Graphs A and C:” and “Graphs B and D:” should be used, according to the rest of the manuscript. In general, this caption is not so clear as the others, the structure should be changed a bit in order to be more consistent.Done
- Line 292: “47, 6%” looks like mistyping.Done
- Line 294: “… differ between IICC and TIFC mosquitoes for …”Done
- Line 300-301: As commented in Line 270-273, it would be better to use the same punctuation in all the captions, without switching between “:” and “-”.Done
- Line 312-315: The caption is a bit messy, since there are much information given in a not so well structured way.Done
- Line 312: I would rephrase: “YFV viral load in head and abdomen of Pakistani Ae. aegypti with kdr …”Done
- Line 314: “mosquito abdomens”.Done
- Line 320: “… and exhibit …”Done
- Line 332: I would rephrase “The similar IR, DR and viral loads observed…”.Done
- Line 336-339: This sentence is substantially repeating the previous one, I would delete it completely, just taking care of correcting the reference (16) in the appropriate way.Done.
- Line 339-343: I would highlight the connection between the conclusion at Line 335 and the beginning of the new sentence about mathematical models, maybe rephrasing like: “Indeed, a mathematical model assessed that the absence of urban yellow fever in Asia cannot be sufficiently explained by vector competence to YFV alone, proposing that other factors, …, may be playing significant roles”.Done
- Line 345-348: What is the aim of the discussion about mutations in arboviruses? Mutations could increase the risk of YFV cycles in Asia? Or is it a generic consideration about arboviruses? I would point this out better.
We rephrased the sentence to provide a more comprehensive idea
- Line 351-354: The point about demographic factors in monkey populations looks disconnected from the discussion on mutations, I would introduce it in a different way.
We revised and expanded the paragraph to provide a more comprehensive explanation and improve the flow of the text
- Line 380: I would use “showed” instead of “presented”.Done
- Line 384: I would use “indeed” instead of “however”, since this sentence is agreeing to what just stated about kdr mutations not affecting the viral infection.Done
- Line 389: I would add something about the absence of influence from kdr on YFV infection, something like: “… is indeed susceptible to YFV infection beside of differences in kdr genotype, indicating…”Done
Once again, thank you very much for these thorough and meticulous reviews.

Reviewer 3 Report
Comments and Suggestions for Authors
Title: Evaluation of Yellow Fever Virus Infection on Aedes aegypti Mosquitoes from Pakistan with Distinct kdr Genotypes
The manuscript tries to shed light on the absence of yellow fever in Asia, despite the presence in the region of the competent vector Aedes aegypti. The study demonstrates that Pakistani Ae. aegypti mosquitoes can be infected with no significant differences compared to a reference American strain. Furthermore, the connection between kdr mutations, causing resistance to pyrethroids, and capability to be infected with yellow fever virus, is investigated, resulting in absence of correlation, differently from previous studies on different mosquitoes and viruses.
The manuscript is well written, and the experiments are reported in a detailed way. However, I noticed a general lack of analysis depth on the results obtained and on the rationale behind the study. It would be wise to highlight why it is important to screen the susceptibility to yellow fever virus infection in mosquitoes with different kdr genotypes, reporting carefully previous cases of increased susceptibility to viruses because of kdr resistance; the finding, in this study, of absence of correlation between kdr and YFV susceptibility is opposite to previous evidence, hence it could be worth trying to understand better the possible reasons of this unexpected result. The study may also look incomplete because of the many questions left open about the reasons of absence of YFV in Asia, which are just described with no proper in-depth analysis. Further considerations in the Discussion could give the manuscript more completeness. I recommend this manuscript for publishing in Insects after revisions.
General comments:
- The logic behind the abstract is fragmented, the connection between kdr and YFV infection should be stated also there, and not only in Introduction and Discussion.
- At the end of the Introduction, the hypothesis of kdr mutations increasing susceptibility to YFV is introduced in a very concise way which could be explained better (also considering that this hypothesis will be refused in the manuscript).
- Most of the figure captions could be rephrased to be easier to read, since the high schematic way they are written can be confusing.
- The Discussion contains a lot of information but lacks analysis depth. Many possible reasons for absence of yellow fever in Asia are given, giving the impression of a list without critical analysis of what could be causing the difference between Asian environment and African and American one.
- The same can be said about the discussion on kdr and susceptibility to YFV, but one important thing is that the mentioned references mostly suggest increased/altered susceptibility to viruses because of kdr mutation, while the previous publication from your group (75) stated that, in Brazilian Ae. aegypti mosquitoes, kdr did not cause any differences in infection with chikungunya virus (hence agreeing with the present manuscript).
- Particular attention should be given to standardize punctuation and structure of phrases; some words and phrasings may be changed.
Detailed comments:
- Line 26-27: the link between yellow fever virus and mutations causing insecticide resistance is not explicated here.
- Line 30: as it is the first mention of YFV, it should be explained: “genetic differences in yellow fever virus (YFV) strains”.
- Line 33: I would add “female adults”.
- Line 53: “sylvatic vectors, such as…”
- Line 53-55: I would rephrase: “the sylvatic cycle, in which the circulation between sylvatic vectors, such as Haemagogus and Sabethes genera, and non-human primates (NHPs) can accidentally infect humans; …”
- Line 56: A comma could be needed: “and the urban cycle, in which humans…”
- Line 61: Maybe: “and all three transmission cycles”.
- Line 61: Rephrase: “with the intermediate one usually serving as a bridge for the urban one”.
- Line 66: “involving immune responses, microbiota and genetic factors, …”
- Line 67: “… and viral genotypes, important determinants…”
- Line 68-69: I would rephrase to give more continuity to the text: “Indeed, one extrinsic factor with a key role for human YFV infections is the presence and distribution of susceptible mosquito vectors, such as Aedes …”
- Line 75-77: Rephrase needed: “… in Asian countries, while other Ae. aegypti arboviruses, such as dengue and chikungunya, are responsible for epidemies in the Asia-Pacific region, as well as in the American and African continents”.
- Line 81-84: I would give a different tone: “Although natural populations (…) and transmission of the virus, laboratory infection assays demonstrated that…”
- Line 91: This part needs to be explained better; it is too stretched.
- Line 95-97: “… for mutations in several Nav sites, both worldwide-spread, as the F1534C substitution, ad regional-specific, as the V410L and V1016I mutations in the Americas and in Africa and the S989P, V1016G and T1520I mutations in Asia”.
- Line 97: I would add: “In Ae. aegypti populations from Pakistan”
- Line 98: I would add: "… forming two different kdr haplotypes:”
- Line 100: This is a crucial point to explain the rationale of the research and I think it should be described in a more detailed way. The idea behind is: since the loss in fitness and the reduction in vector capacity have already been studied for several mosquitoes and viruses, but no studies have been reported for Ae. aegypti infections with YFV, it would be important to check if kdr mosquitoes have enhanced susceptibility to this virus. This concept should appear clearer, without jumping to the Discussion or reading the mentioned references.
- Line 109: “… as colonies in insectaries”.
- Line 115: in “2L”, it would be better to separate the number from the measure unit, since in the next lines they are always separated.
- Line 129: better to add a comma before “isolated”.
- Line 134: “24-hours”.
- Line 141: need to explain “B.O.D.”.
- Line 146: “Mosquitoes”.
- Line 153: “… as described by the…”
- Line 156: “… instructions; the nucleic acid was stored…”
- Line 156: I would speak about “reactions” as plural.
- Line 174-175: “Positions are indicated in relation to...”
- Line 189: “Ae. aegypti”.
- Line 196: I would say: “… temperatures, allowing for qPCR detection…”
- Line 205: Why are the degrees symbols here looking different compared to other parts of the manuscript?
- Line 214: I would use “used” instead of “utilized”.
- Line 247-248: “Graph” should be written as plurals.
- Line 249-250: I would rephrase: “Our findings of YFV IR and DR were compared with previous records for Asian Ae. aegypti populations (Figure 2)”.
- Line 270-273: I would suggest keeping using the same punctuation: “Graphs A and C:” and “Graphs B and D:” should be used, according to the rest of the manuscript. In general, this caption is not so clear as the others, the structure should be changed a bit in order to be more consistent.
- Line 292: “47, 6%” looks like mistyping.
- Line 294: “… differ between IICC and TIFC mosquitoes for …”
- Line 300-301: As commented in Line 270-273, it would be better to use the same punctuation in all the captions, without switching between “:” and “-”.
- Line 312-315: The caption is a bit messy, since there are much information given in a not so well structured way.
- Line 312: I would rephrase: “YFV viral load in head and abdomen of Pakistani Ae. aegypti with kdr …”
- Line 314: “mosquito abdomens”.
- Line 320: “… and exhibit …”
- Line 332: I would rephrase “The similar IR, DR and viral loads observed…”.
- Line 336-339: This sentence is substantially repeating the previous one, I would delete it completely, just taking care of correcting the reference (16) in the appropriate way.
- Line 339-343: I would highlight the connection between the conclusion at Line 335 and the beginning of the new sentence about mathematical models, maybe rephrasing like: “Indeed, a mathematical model assessed that the absence of urban yellow fever in Asia cannot be sufficiently explained by vector competence to YFV alone, proposing that other factors, …, may be playing significant roles”.
- Line 345-348: What is the aim of the discussion about mutations in arboviruses? Mutations could increase the risk of YFV cycles in Asia? Or is it a generic consideration about arboviruses? I would point this out better.
- Line 351-354: The point about demographic factors in monkey populations looks disconnected from the discussion on mutations, I would introduce it in a different way.
- Line 380: I would use “showed” instead of “presented”.
- Line 384: I would use “indeed” instead of “however”, since this sentence is agreeing to what just stated about kdr mutations not affecting the viral infection.
- Line 389: I would add something about the absence of influence from kdr on YFV infection, something like: “… is indeed susceptible to YFV infection beside of differences in kdr genotype, indicating…”
Author Response
Reviewer 3
The manuscript tries to shed light on the absence of yellow fever in Asia, despite the presence in the region of the competent vector Aedes aegypti. The study demonstrates that Pakistani Ae. aegypti mosquitoes can be infected with no significant differences compared to a reference American strain. Furthermore, the connection between kdr mutations, causing resistance to pyrethroids, and capability to be infected with yellow fever virus, is investigated, resulting in absence of correlation, differently from previous studies on different mosquitoes and viruses.
The manuscript is well written, and the experiments are reported in a detailed way. However, I noticed a general lack of analysis depth on the results obtained and on the rationale behind the study. It would be wise to highlight why it is important to screen the susceptibility to yellow fever virus infection in mosquitoes with different kdr genotypes, reporting carefully previous cases of increased susceptibility to viruses because of kdr resistance; the finding, in this study, of absence of correlation between kdr and YFV susceptibility is opposite to previous evidence, hence it could be worth trying to understand better the possible reasons of this unexpected result. The study may also look incomplete because of the many questions left open about the reasons of absence of YFV in Asia, which are just described with no proper in-depth analysis. Further considerations in the Discussion could give the manuscript more completeness. I recommend this manuscript for publishing in Insects after revisions.
>>We thank “Reviewer 3” for their meticulous review and encouraging feedback. We have refined the text to better highlight the rationale behind our hypothesis: that an Asian Ae. aegypti population can be efficiently infected with distinct YFV strains, regardless of its pyrethroid resistance genotype (in this case, associated with kdr mutations), which are often suggested to strongly influence various physiological traits in the organism. Given our findings of no significant differences, we emphasize that factors beyond the mosquito genetic background and virus strains may be contributing to the absence of yellow fever transmission in Asia.
General comments:
- The logic behind the abstract is fragmented, the connection between kdr and YFV infection should be stated also there, and not only in Introduction and Discussion.
>> Agreed. We have revised the abstract to address this comment.
- At the end of the Introduction, the hypothesis of kdr mutations increasing susceptibility to YFV is introduced in a very concise way which could be explained better (also considering that this hypothesis will be refused in the manuscript).
>>This paragraph has been revised to clarify our hypothesis.
- Most of the figure captions could be rephrased to be easier to read, since the high schematic way they are written can be confusing.
>>We revised all the figures, including the “Reviewer 1” concerns.
- The Discussion contains a lot of information but lacks analysis depth. Many possible reasons for absence of yellow fever in Asia are given, giving the impression of a list without critical analysis of what could be causing the difference between Asian environment and African and American one.
>> We revised the Discussion section considering this advice.
- The same can be said about the discussion on kdr and susceptibility to YFV, but one important thing is that the mentioned references mostly suggest increased/altered susceptibility to viruses because of kdr mutation, while the previous publication from your group (75) stated that, in Brazilian Ae. aegypti mosquitoes, kdr did not cause any differences in infection with chikungunya virus (hence agreeing with the present manuscript).
>>Yes, we agree. In the revised text we highlighted these findings.
- Particular attention should be given to standardising punctuation and structure of phrases; some words and phrasings may be changed.
>> We appreciate this reminder. We also noticed that our original text was edited to fit the journal’s format, which altered some of our sentences. We have corrected these in addition to addressing the reviewers’ comments.
Detailed comments:
- Line 26-27: the link between yellow fever virus and mutations causing insecticide resistance is not explicated here.
Corrected
- Line 30: as it is the first mention of YFV, it should be explained: “genetic differences in yellow fever virus (YFV) strains”.Done
- Line 33: I would add “female adults”.Done
- Line 53: “sylvatic vectors, such as…”Done
- Line 53-55: I would rephrase: “the sylvatic cycle, in which the circulation between sylvatic vectors, such as Haemagogus and Sabethes genera, and non-human primates (NHPs) can accidentally infect humans; …”Done
- Line 56: A comma could be needed: “and the urban cycle, in which humans…”Done
- Line 61: Maybe: “and all three transmission cycles”.Done
- Line 61: Rephrase: “with the intermediate one usually serving as a bridge for the urban one”.Done
- Line 66: “involving immune responses, microbiota and genetic factors, …”Done
- Line 67: “… and viral genotypes, important determinants…”Done
- Line 68-69: I would rephrase to give more continuity to the text: “Indeed, one extrinsic factor with a key role for human YFV infections is the presence and distribution of susceptible mosquito vectors, such as Aedes …”Done
- Line 75-77: Rephrase needed: “… in Asian countries, while other Ae. aegypti arboviruses, such as dengue and chikungunya, are responsible for epidemies in the Asia-Pacific region, as well as in the American and African continents”Done.
- Line 81-84: I would give a different tone: “Although natural populations (…) and transmission of the virus, laboratory infection assays demonstrated that…”Done
- Line 91: This part needs to be explained better; it is too stretched.Done
- Line 95-97: “… for mutations in several Nav sites, both worldwide-spread, as the F1534C substitution, ad regional-specific, as the V410L and V1016I mutations in the Americas and in Africa and the S989P, V1016G and T1520I mutations in Asia”.Done
- Line 97: I would add: “In Ae. aegypti populations from Pakistan”Done
- Line 98: I would add: "… forming two different kdr haplotypes:”Done
- Line 100: This is a crucial point to explain the rationale of the research and I think it should be described in a more detailed way. The idea behind is: since the loss in fitness and the reduction in vector capacity have already been studied for several mosquitoes and viruses, but no studies have been reported for Ae. aegypti infections with YFV, it would be important to check if kdr mosquitoes have enhanced susceptibility to this virus. This concept should appear clearer, without jumping to the Discussion or reading the mentioned references.
>> Thank you.We expanded this idea in the paragraph and added the corresponding references
- Line 109: “… as colonies in insectaries”.Done
- Line 115: in “2L”, it would be better to separate the number from the measure unit, since in the next lines they are always separated.Done
- Line 129: better to add a comma before “isolated”.Done
- Line 134: “24-hours”.Done
- Line 141: need to explain “B.O.D.”.Done
- Line 146: “Mosquitoes”.Done
- Line 153: “… as described by the…”Done
- Line 156: “… instructions; the nucleic acid was stored…”Done
- Line 156: I would speak about “reactions” as plural.Done
- Line 174-175: “Positions are indicated in relation to...”Done
- Line 189: “Ae. aegypti”.Done
- Line 196: I would say: “… temperatures, allowing for qPCR detection…”Done
- Line 205: Why are the degrees symbols here looking different compared to other parts of the manuscript?It was a different pattern. We corrected this issue
- Line 214: I would use “used” instead of “utilized”.Done
- Line 247-248: “Graph” should be written as plurals.Done
- Line 249-250: I would rephrase: “Our findings of YFV IR and DR were compared with previous records for Asian Ae. aegypti populations (Figure 2)”.Done
- Line 270-273: I would suggest keeping using the same punctuation: “Graphs A and C:” and “Graphs B and D:” should be used, according to the rest of the manuscript. In general, this caption is not so clear as the others, the structure should be changed a bit in order to be more consistent.Done
- Line 292: “47, 6%” looks like mistyping.Done
- Line 294: “… differ between IICC and TIFC mosquitoes for …”Done
- Line 300-301: As commented in Line 270-273, it would be better to use the same punctuation in all the captions, without switching between “:” and “-”.Done
- Line 312-315: The caption is a bit messy, since there are much information given in a not so well structured way.Done
- Line 312: I would rephrase: “YFV viral load in head and abdomen of Pakistani Ae. aegypti with kdr …”Done
- Line 314: “mosquito abdomens”.Done
- Line 320: “… and exhibit …”Done
- Line 332: I would rephrase “The similar IR, DR and viral loads observed…”.Done
- Line 336-339: This sentence is substantially repeating the previous one, I would delete it completely, just taking care of correcting the reference (16) in the appropriate way.Done.
- Line 339-343: I would highlight the connection between the conclusion at Line 335 and the beginning of the new sentence about mathematical models, maybe rephrasing like: “Indeed, a mathematical model assessed that the absence of urban yellow fever in Asia cannot be sufficiently explained by vector competence to YFV alone, proposing that other factors, …, may be playing significant roles”.Done
- Line 345-348: What is the aim of the discussion about mutations in arboviruses? Mutations could increase the risk of YFV cycles in Asia? Or is it a generic consideration about arboviruses? I would point this out better.
We rephrased the sentence to provide a more comprehensive idea
- Line 351-354: The point about demographic factors in monkey populations looks disconnected from the discussion on mutations, I would introduce it in a different way.
We revised and expanded the paragraph to provide a more comprehensive explanation and improve the flow of the text
- Line 380: I would use “showed” instead of “presented”.Done
- Line 384: I would use “indeed” instead of “however”, since this sentence is agreeing to what just stated about kdr mutations not affecting the viral infection.Done
- Line 389: I would add something about the absence of influence from kdr on YFV infection, something like: “… is indeed susceptible to YFV infection beside of differences in kdr genotype, indicating…”Done
Once again, thank you very much for these thorough and meticulous reviews.
